# Screening of a *Haloferax volcanii* Transposon Library Reveals Novel Motility and Adhesion Mutants

**DOI:** 10.3390/life6040041

**Published:** 2016-11-26

**Authors:** Georgio Legerme, Evan Yang, Rianne N. Esquivel, Saija Kiljunen, Harri Savilahti, Mechthild Pohlschroder

**Affiliations:** 1Department of Biology, University of Pennsylvania, Philadelphia 19104, PA, USA; gplegerme@gmail.com (G.L.); evanyang@sas.upenn.edu (E.Y.); rianne.esquivel@gmail.com (R.N.E.); 2Division of Genetics and Physiology, Department of Biology, University of Turku, Turku 20500, Finland; saija.kiljunen@helsinki.fi (S.K.); harri.savilahti@utu.fi (H.S.); 3Immunobiology Program, Research Programs Unit, University of Helsinki, Fi-00014, Helsinki, Finland

**Keywords:** *Haloferax volcanii*, transposon mutagenesis, archaeon, flagellum, archaellum, type IV pilus, adhesion, biofilm, swimming motility, chemotaxis

## Abstract

Archaea, like bacteria, use type IV pili to facilitate surface adhesion. Moreover, archaeal flagella—structures required for motility—share a common ancestry with type IV pili. While the characterization of archaeal homologs of bacterial type IV pilus biosynthesis components has revealed important aspects of flagellum and pilus biosynthesis and the mechanisms regulating motility and adhesion in archaea, many questions remain. Therefore, we screened a *Haloferax volcanii* transposon insertion library for motility mutants using motility plates and adhesion mutants, using an adapted air–liquid interface assay. Here, we identify 20 genes, previously unknown to affect motility or adhesion. These genes include potential novel regulatory genes that will help to unravel the mechanisms underpinning these processes. Both screens also identified distinct insertions within the genomic region lying between two chemotaxis genes, suggesting that chemotaxis not only plays a role in archaeal motility, but also in adhesion. Studying these genes, as well as hypothetical genes *hvo_2512* and *hvo_2876*—also critical for both motility and adhesion—will likely elucidate how these two systems interact. Furthermore, this study underscores the usefulness of the transposon library to screen other archaeal cellular processes for specific phenotypic defects.

## 1. Introduction

In archaea, similar to bacteria, the ability to move and adhere to surfaces is an essential part of cell life, aiding in processes such as movement toward nutrients and away from toxins, as well as resisting environmental stressors through the establishment of biofilms [1]. Certain aspects of these archaeal functions and the biology responsible for them resemble those found in bacteria, such as the dependence of many archaea on rotating flagella (archaella) for swimming motility, and type IV pili for surface adhesion [2,3]. However, many aspects of these processes are unique to archaea. Therefore, strategies such as genetic selections are imperative to further elucidate the mechanisms underpinning archaeal movement and surface adhesion, an early step in biofilm formation.

Although type IV pilus biosynthesis was found to be conserved between bacteria and archaea, the archaeal flagellum and its subunits, the flagellins, share no homology with their bacterial counterparts [3]. Rather, their biosynthesis components share homology with those of the type IV pilus biosynthesis machinery, suggesting a functional intersection between these two systems. For example, the archaeal flagellins, despite lacking homology to pilins—the pilus subunits—have a signal peptide structure that is similar to that of the pilin signal peptides. Both signal peptides are recognized and processed by a signal peptidase, PibD, a homolog of the bacterial PilD [4,5]. Unlike the universally conserved signal peptidase I, which removes the entire Sec signal peptide including the hydrophobic (H) domain, PibD processing retains the pilin and flagellin H-domains, which form the core of the pilus and flagellum, respectively [6,7]. Additionally, just as type IV pilus biosynthesis requires the ATPase, PilB, and the membrane protein, PilC, flagella biosynthesis also requires the PilB homolog, FlaI, as well as the PilC homolog, FlaJ [3,8]. Interestingly, FlaI contains a second ATPase active site, which is required for flagella rotation [9]. Flagella rotation in a halophilic archaeon, *Halobacterium salinarum* (*Hbt*), was shown to require FlaC, FlaD and FlaE, which use archaea-specific chemotaxis (Che) proteins to interact with homologs of the bacterial chemotaxis signaling system, such as CheY, a response regulator, and CheD, a receptor activator [10]. However, unlike in Euryarchaea, little is known about chemotaxis in members of the kingdom Crenarchaeota. While crenarchaeota contain a FlaI-interacting ring-like scaffold forming protein, FlaX, that shares some similarity with methyl-accepting chemotaxis proteins, they lack proteins with significant similarity to the aforementioned euryarchaeal Fla proteins or any known chemotaxis components [11,12].

Conversely, several components of the crenarchaeal flagellum regulatory network (Arn) involved in regulation of the flagella operon have been identified and characterized [13,14], while the first euryarchaeal flagellum transcription regulator (EarA) has only recently been identified in *Methanococcus maripaludis* [15]. Similarly, while three crenarchaeal biofilm regulators (AbfRs) were established to influence biofilm formation, regulation of biofilm growth in euryarchaea is less understood [16]. However, we have recently shown that pilins are involved in regulating flagella biosynthesis in the euryarchaeon, *Haloferax volcanii* (*Hfx*), likely by sequestering an as-yet unidentified protein, or proteins, in the membrane [17]. When these pilins are incorporated into pili, this sequestered factor is released and inhibits motility. *N*-glycosylation also appears to play roles in regulating both motility and the early steps of biofilm formation [18,19]. Both the pilins and the flagellins can be modified by *N*-glycosylation, but interestingly, while flagella-dependent motility is inhibited in cells that are defective for *N*-glycosylation, certain pilins switch their functions depending on their glycosylation state [19]. However, not all components of this archaeal glycosylation pathway, or of the other recently characterized glycosylation pathways, have been identified [20].

With much left to be discovered about the structure, biosynthesis, and the regulation involved with archaeal motility and adhesion, we screened a recently generated transposon insertion library for mutants with altered motility or adhesion characteristics [21]. Mutants with either motility or adhesion defects were identified using motility assay plates with a low agar concentration and a modified 96-well plate air–liquid interface (ALI) adhesion assay, respectively. We found several interesting mutants, including some harboring insertions in genes known to be involved in at least one of these processes, as well as novel motility and adhesion mutants. Some of these mutants exhibited either inhibited adhesion or defective motility, while others exhibited defects in both of these characteristics. The identification of genes that were not previously known to be involved in swimming motility and/or surface adhesion strongly suggests that we are only beginning to understand the mechanisms of how basic functions such as motility and adhesion operate in *Hfx. volcanii*.

## 2. Materials and Methods

**Reagents.** All enzymes used for standard molecular biology procedures were purchased from New England BioLabs (Ipswich, MA, USA), except for iProof High-Fidelity DNA polymerase, which was purchased from Bio-Rad (Hercules, CA, USA). The Enhanced Chemiluminescence (ECL) Plus Western blotting detection system and horseradish peroxidase-linked sheep anti-mouse antibodies were purchased from Amersham Biosciences (Little Chalfont, UK). The Polyvinylidene difluoride (PVDF) membrane was purchased from Millipore (Billerica, MA, USA). DNA and plasmid purification kits and anti-His antibodies were purchased from Qiagen (Hilden, Germany). NuPAGE gels, buffers, and reagents were purchased from Invitrogen (Carlsbad, CA, USA). Difco agar and Bacto yeast extract were purchased from Becton Dickinson (Franklin Lakes, NJ, USA). Peptone was purchased from Oxoid (Altrincham, UK) and 5-fluoroorotic acid (5-FOA) was purchased from Toronto Research Biochemicals (Toronto, ON, Canada). All other chemicals and reagents were purchased from either Thermo Fisher Scientific (Waltham, MA, USA) or Sigma-Aldrich (St. Louis, MO, USA). 

**Strains and growth conditions.** The plasmids and strains used in this study are listed in Table 1. *Hfx. volcanii* strain H53 and its derivatives were grown at 45 °C in liquid, solid, or semi-solid semi-defined casamino acid (CA) medium supplemented with tryptophan and uracil (50 µg mL^−1^ final concentration) [22]. Solid medium plates contained 1.5% agar, whereas motility assay plates contained 0.3% agar. To ensure equal agar concentrations in all plates, agar was completely dissolved in the medium prior to autoclaving, and the autoclaved medium was stirred before being poured into plates. Strains transformed with pTA963 were grown on CA medium supplemented with tryptophan (50 µg mL^−1^ final concentration). FH37, an H53 strain expressing a chromosomally encoded tryptophan gene located downstream of the S-layer glycoprotein encoding gene, and transposon mutants were grown on CA medium supplemented with uracil (50 µg mL^−1^ final concentration). For selection of the deletion mutant, 5-FOA was added at a final concentration of 150 µg mL^−1^ in CA medium and uracil was added to a final concentration of 10 μg mL^−1^. *Escherichia coli* strains were grown at 37 °C in NZCYM (NZ-amine, Casamino Acid, Yeast Extract, MgSO4) medium supplemented with ampicillin (200 µg mL^−1^) [23]. NZCYM was obtained from Thermo Fisher Scientific (10 g/L tryptone, 5 g/L yeast extract, 1 g/L casamino acids, 5 g/L NaCl, 1 g/L MgSO_4_).

***Hfx. volcanii* transposon insertion library.** To multiply the original *Hfx. volcanii* transposon insertion library described in [21], the library was plated out onto Hv-CA plates supplemented with uracil (50 µg mL^−1^) and cultured for 7 days. Altogether, 24,000 colonies were collected in a total volume of 108 mL Hv-YPC [22] medium containing 20% glycerol that was divided into 800 µL aliquots, frozen using liquid nitrogen and stored at −75 °C.

**Motility assay.** Motility assays were performed on plates containing 0.3% agar in CA medium supplemented with uracil (50 µg mL^−1^ final concentration). Colonies from the transposon mutant library were stab-inoculated into the agar, and the halo size around each inoculation site was measured periodically [29].

**Surface adhesion assay.**
*Hfx. volcanii* surface adhesion to a plastic cover slip was assayed using a modified ALI assay as described by Esquivel et al. [26,30]. Briefly, 3 mL of culture in CA medium supplemented with uracil was grown to an optical density at 600 nm (OD_600_) of ~0.3 and incubated in each well of a covered 12-well plate. Plastic cover slips (22 × 22 mm; 0.19–0.25 mm thick) from Fisher Scientific (Hampton, NH, USA) were inserted into each well at a 90° angle and incubated overnight at 45 °C without shaking. After 2% (*v/v*) acetic acid fixation for 3 min., coverslips were air-dried and then stained in 0.1% (*w/v*) crystal violet solution for 10 min. The coverslips were then washed with distilled water, air-dried, and examined using light microscopy.

Surface adhesion to 96-well plates was assayed through a protocol developed here, modifying the ALI-assay described by O’Toole [30]. Briefly, 200 μL of semi-defined CA medium supplemented with tryptophan was inoculated, and cells were grown at 45 °C with shaking. However, to promote surface adhesion without impeding growth, short periods of static incubation (4 min during each 20 min interval) were also used along with an 8 h period of static incubation at the end of 32 h for a total incubation period of 40 h. Following the 40 h incubation, culture media was discarded and cells adhering to the wells were fixed with 2% (*v/v*) acetic acid for 3 min, allowed to air dry, stained with 0.1% (*w/v*) crystal violet for 10 min, and rinsed three times with water. To quantify cell surface adhesion, 200 µL of a methanol–acetic acid mix (10% acetic acid, 30% methanol) was used to release the crystal violet stain from the adhering cells, and the optical density of each sample was determined at 600 nm (OD_600_) using a Biotek reader (BioTek Instruments, Winooski, VT, USA).

**Generation of chromosomal deletion.** A chromosomal deletion of *hvo_2876* was generated in the FH37 wild-type strain using a homologous-recombination-based (pop-in pop-out) method [31]. Plasmid constructs were generated using overlap PCR [29] with the primers listed in Table 2. To confirm that the chromosomal replacement event occurred at the proper location on the chromosome, genomic DNA isolated from colonies derived using these techniques was screened by PCR using two primer pairs, one hybridizing within the deletion and the other hybridizing to genomic sequences flanking the deletion (Table 2). The identities of the PCR products were verified by DNA sequencing.

**Western blot analysis.** Proteins from cell pellets of *Hfx. volcanii* strains containing His-tagged constructs were separated by gel electrophoresis in a lithium dodecyl sulfate (LDS) sample buffer and analyzed by Western blot using anti-His antibodies. Separation using LDS had greater resolution and decreased band smearing than those conducted with sodium dodecyl sulfate (SDS) when using the NuPAGE system (Novex, San Diego, CA, USA). Liquid cultures were grown to early log phase (OD_600_ ~0.3). Cells were collected by centrifugation at 13,000× rpm for 3 min. Supernatants were collected and proteins were precipitated with 100% trichloroacetic acid (TCA). Proteins were centrifuged at 13,000× rpm for 20 min and washed with 80% acetone at 4 °C. Proteins from cell and supernatant fractions were resuspended in 1% (*v/v*) NuPAGE LDS supplemented with 50 mM dithiothreitol. The electrophoresis of protein samples was performed with 12% (*v/v*) BisTris NuPAGE gels under denaturing conditions using 1× MOPS buffer (pH 7.7). Proteins were electroeluted from the gel onto a PVDF membrane using a Transblot-SD semidry transfer cell (Bio-Rad) at 15 V for 30 min. Western blots of samples expressing C-terminally His-tagged constructs with a three-amino acid linker sequence were probed with an anti-penta·His antibody (Qiagen) at a dilution of 1:2000 followed by a secondary anti-mouse antibody (Amersham Biosciences) at a dilution of 1:10,000. Antibody-labeled protein bands were identified using the Amersham Biosciences ECL Plus Western blotting detection system.

**Transmission Electron Microscopy (TEM).**
*Hfx. volcanii* whole cells were prepared as previously described [32]. Cell cultures at an OD_600_ of ~0.3 in CA medium were fixed with 2% (*v/v*) glutaraldehyde and 1% (*v/v*) paraformaldehyde for 1 h. An amount of 10 µL of the fixed culture was put onto glow-discharged copper grids coated with carbon-Formvar for 10 min. The grids were rinsed two times in ddH_2_O and negatively stained using 1% (*w/v*) uranyl formate. Grids were then analyzed using the JEOL10 transmission electron microscope (JEOL Ltd.; Akishima, Tokyo, Japan).

**Identification of transposon integration sites by next-generation sequencing.** A Nextera Kit (Illumina; San Diego, CA, USA) was used to prepare the motility and/or adhesion mutant DNA for genome sequencing using a 300 cycle micro v2 MiSeq (Illumina). DNA was purified with the Zymo DNA clean and concentrator Kit (Irvine, CA, USA) and diluted in sterile water to a concentration of 10 ng/µL. Mutant genomic reads were assembled and mapped to the reference transposon sequence using the program, Geneious (Biomatters Ltd; Auckland, New Zealand), and the genomic sequence flanking the transposon was used to identify the transposon-disrupted gene through the National Center for Biotechnology Information’s (NCBI) Basic Local Alignment Search Tool (BLAST). Gene annotations were obtained from NCBI unless otherwise noted.

## 3. Results and Discussion

### 3.1. Isolation and Identification of Motility Mutants from a Hfx. volcanii Transposon Insertion Library

To identify non-motile transposon mutants, individual colonies from the *Hfx. volcanii* transposon mutant library were stab-inoculated into motility agar plates (Figure 1). Of the 8800 mutants screened, 17 mutants were non-motile even after five transfers and incubation on a motility plate for up to 10 days. Mutants that had reduced motility were not considered for this study. None of the mutants identified and included in this study exhibited a noticeable growth defect when grown on a solid agar plate or in liquid medium (data not shown).

From these 17 mutants, we identified the transposon insertion site using whole-genome sequencing (Table 3). Each insertion was unique and helped to identify 11 genes and one significant genomic region, of which two genes and the genomic region were hit twice and one gene was hit three times. Only four of the insertions were located in genes or genomic regions previously known to be involved in motility.

Consistent with our screen successfully identifying non-motile mutants, two independent mutants contained an insertion in *pibD*, the gene that encodes the preflagellin and prepilin peptidase [29]. Furthermore, in accordance with previous studies showing that mutations in genes required for chemotaxis also affect the motility of the haloarchaeon, *Hbt. salinarum*, two distinct mutants were identified as having insertions in the genomic region containing *Hfx. volcanii* chemotaxis-related genes (*che* genes). The respective transposon insertions were located between *cheB* and *cheW1*, genes that are transcribed in opposite directions. *CheB* and *cheW1* encode a protein-glutamate methylesterase and purine-binding protein, respectively, proteins that have been shown to be involved in motility in both bacteria and the haloarchaeon, *Hbt. salinarum* [33,34].

The majority of non-motile mutants had a transposon inserted in genes that were not previously known to be involved in motility (Table 3). Some of these genes might encode proteins that play a role in regulating motility, such as *hvo_0246*, which encodes a putative ArsR family regulator protein. Two motility deficient mutants were identified that had independent and distinct insertions in this particular gene. Putative functions of the proteins encoded by the other disrupted genes identified in this screen are known. These include *hvo_1098*, encoding a diaminopimelate decarboxylase; *hvo_0069*, encoding an arylsulfatase; and *hvo_3002*, encoding an ABC transporter ATP-binding protein. However, while these functions are known in a general sense, the specific roles that these proteins play in motility are uncertain. Interestingly, three of these motility mutants contain insertions within *hvo_3001*, which lies downstream of *hvo_3002* and encodes an ABC transporter permease. While both genes encode proteins that might be important for metal transport with the closest related genes of known function encoding proteins that play roles in copper uptake systems, it is possible that the motility defect caused by the transposon insertion in *hvo_3002* is due to a downstream effect on *hvo_3001* expression. Another gene, *hvo_2876*, encodes a hypothetical protein that is 22 bp upstream of a gene encoding a methyltransferase-like protein transcribed in the same direction, suggesting that they are components of an operon.

Adhesion assays revealed, as had been shown in previous studies with a *pibD* deletion mutant, that the *Hfx. volcanii pibD::tn* mutants are also unable to adhere to plastic cover slips immersed in a mid-log culture for 12 h (Figure 2) [29]. However, less expected, *Hfx. volcanii* mutants having a transposon insertion between *cheB* and *cheW1* also exhibited phenotypic defects in surface adhesion (Figure 2) strongly suggesting that, in *Hfx. volcanii*, chemotaxis plays an important role in both adhesion and motility. Although the connection between these chemotaxis genes and motility had already been established in archaea [10,34,35,36], to the best of our knowledge, this is the first indication in any archaeon that chemotaxis is also required for adhesion. This is particularly intriguing since we have previously shown that, unlike for bacterial adhesion, flagella-dependent motility is not required for *Hfx. volcanii* adhesion [29]. Hence, these mutants provide an opportunity to further understand how the motility and adhesion biological systems interact.

Disruption of *hvo_2876*, which encodes a hypothetical protein of unknown function, also caused an adhesion defect (Figure 2). Since, like Δ*pibD*, *hvo_2876::tn*, almost completely lost the ability to adhere, it was selected for further characterization.

### 3.2. Generation and Characterization of Δ*hvo_2876*

To determine whether the motility and adhesion phenotypes of *hvo_2876::tn* were indeed due to the disrupted expression of this hypothetical protein, we generated a chromosomal deletion of *hvo_2876* (see Materials and Methods) (Figure 3A,B). Phenotypic characterization of Δ*hvo_2876* revealed that this deletion strain is indeed both non-motile and non-adhering, consistent with the transposon mutant phenotypes (Figure 2 and Figure 3C,D).

Transmission electron microscopy of both *hvo_2876::tn* and Δ*hvo_2876* strains revealed similar filamentation phenotypes (Figure 4). While surface filaments could be observed in both strains, fewer cells had filaments compared to the wild-type strain. Likewise, while wild-type cells generally have 1–12 long filaments [19], none of the *hvo_2876::tn* or Δ*hvo_2876* cells examined had more than one long filament. Intriguingly, these cells had a significant number of short filamentous surface structures. However, it cannot be excluded that these structures might be the result of cell lysis.

Since flagella and pili have similar diameters, and we do not have antibodies against their subunits, they cannot be distinguished by TEM. Therefore, to determine whether *hvo_2876* mutants produce stable flagella and/or pili, we expressed C-terminally His-tagged major flagella and pili subunits, FlgA1His and PilA2His, respectively, in wild-type and Δ*hvo_2876* strains. Using anti-His antibodies, Western blot analyses were performed on the cell and supernatant fractions of these strains. Since pili and flagella are sheered from the cells during centrifugation, His-tagged flagellins and pilins in the supernatant fraction would suggest biosynthesis of flagella and pili, respectively, in these cells. When we transform a strain lacking *aglB*, which encodes an oligosaccharyltransferase important for flagellin N-glycosylation, Western blot analysis does not detect a band in the culture supernatant. This lack of *N-*glycosylation results in either a lack of flagella biosynthesis or unstable flagella (Figure 5) [18]. In contrast, Western blot analysis of the supernatant fraction of a culture of the Δ*hvo_2876* strain does detect a faint band (Figure 5A). This result indicates that *hvo_2876* is not required for flagella biosynthesis. However, the significantly lower ratio of flagella in the supernatant compared to the cell fraction is consistent with a lower rate of flagellation in this mutant strain (Figure 4). Faster migrating bands in wild-type and mutant strains are likely due to protein degradation.

Similar results were observed for Western blot analyses of cell and supernatant fractions of wild-type and Δ*hvo_2876* cultures expressing PilA2His indicating reduced, but not absent, piliation in the mutant strain (Figure 5B). As previously determined, unlike flagellation, PilA2 piliation does not require AglB-dependent glycosylation. Hence, while not prominent, PilA2His pilins expressed in a Δ*aglB* Δ*pilA1–6* strain can be observed in the supernatant. Consistent with the lack of AglB-dependent *N-glycosylation*, PilA2His isolated from this strain migrates faster in PAGE than the same protein isolated from wild-type supernatant fractions. FlgA1His and PilA2His expressed in a Δ*hvo_2876* strain migrated similar to these proteins expressed in a wild-type strain. However, this does not exclude the possibility that these proteins, when expressed in this mutant strain, lack some small modification such as methylation, considering that *hvo_2875* encodes a methyltransferase.

### 3.3. Modification of the 96-Well ALI Assay for Use in Screening a Hfx. volcanii Transposon Insertion Library

As described above, the motility screen also identified novel adhesion components (Figure 2). To identify additional surface adhesion mutants, we used a modified 96-well air–liquid interface (ALI) assay. The ALI assay is used to observe adhesion and biofilm formation phenotypes [30]. Since air-borne oxygen is the only electron acceptor available to cells maintained in 96-well plates, they tend to accumulate at the air–liquid interface. By fixing and staining the cells that form a ring on the well surface at this interface, we can determine the relative efficiency with which cells adhere. Cells harboring mutations that cause a significant decrease in the strength of surface adhesion can be readily identified since they will form rings that stain less intensely. This strategy has been successfully employed for a large and diverse array of bacterial species, including *Pseudomonas aeruginosa*, *Vibrio cholerae*, and *E. coli*, and led to the identification of the roles that type IV pili, as well as other proteins and protein structures, play in biofilm formation [30]. However, optimization for use with archaeal species had not yet been established. We have now adapted this assay for the model archaeon, *Hfx. volcanii*, using 96-well plates that allow for large-scale screens. To differentiate between normal and deficient adhesion, FH37 as well as a ∆*pilA[1–6]* strain, which is adhesion deficient due to a lack of major pilins, were used as standards in these assays [26]. FH37 was compared to H53 to show that *trpA* expression from the transposons inserted in the adhesion mutant genomes was not the cause of the adhesion defects observed. Since both adhere similarly when grown in CA+Ura and CA+Ura+Trp liquid media, respectively, it allowed us to grow all strains in CA+Ura liquid media for future adhesion assays. Wild-type surface adhesion was determined to be about double that of the Δ*pilA[1–6]* mutant, with no growth defects observed for either strain (Figure 6). These results correspond well with expectations, strongly suggesting that this modified assay can be used to determine surface adhesion phenotypes of *Hfx. volcanii* strains.

### 3.4. Identification and Characterization of a Genetically Diverse Set of Adhesion Deficient Mutants

By screening the transposon insertion mutant library generated for *Hfx. volcanii* using the aforementioned modified 96-well ALI assay, about 1100 transposon mutants were examined for adhesion defects. Using this approach, we detected and isolated 11 transposon mutants that exhibited reduced adhesion phenotypes but have no growth defects (Figure 7). Mutants with an apparent adhesion defect and normal growth rate were then assessed for their ability to adhere to a plastic cover slip immersed overnight in mutant cultures in 12-well plates (Figure 8). Only adhesion mutants that also showed a significant adhesion defect under these conditions were further considered in this study. As with the motility mutants, we used whole-genome sequencing to identify the transposon insertion sites in the adhesion mutants (Table 4).

Like in the motility screen, we identified one adhesion mutant in which the transposon insertion occurred between *cheB* and *cheW1* in a location distinct from that of the aforementioned motility mutants with transposon insertions between these two chemotaxis genes (Table 3 and Table 4). This result not only further supports the notion that chemotaxis is indeed involved in facilitating *Hfx. volcanii* adhesion, but also confirms the efficacy of the 96-well screen as a means of identifying adhesion mutants.

Given that we know little about how pilus biosynthesis is regulated, it is interesting to note that many of the mutants identified had insertions in genes involved in transcriptional regulation or two-component regulatory systems. This may provide additional opportunities for further studies aimed at determining the specific types of mechanisms that regulate surface adhesion. For example, the *trh1::tn* insertion may disrupt the expression of a transcriptional regulator of the Lrp/AsnC (leucine-responsive regulatory protein/asparagine synthase C) protein family; in bacteria this protein family has been linked to the regulation of pilus biosynthesis, metabolism, and persistence under stress conditions [37]. Another mutant was found to contain a disrupted gene with an insertion in *hvo_0169* encoding a sugar-specific transcriptional regulator, TrmB. In the thermophilic archaeal species, *Pyrococcus furiosus*, as well as *Thermoccus litoralis*, TrmB was found to repress genes encoding the trehalose/maltose ABC transporter [38]. Interestingly, one transposon mutant was found in *hvo_0567* with an insertion site less than 30 nucleotides away from another gene encoding TrmB [*hvo_0568*]. While the adhesion defect of the *hvo_0567* transposon mutant may be due to downstream effects on the TrmB-encoding *hvo_0568*, it is curious that both Hvo_0169 and Hvo_0567, an alpha-amylase whose function is to break down amylose into glucose and maltose, appear to be involved in regulating and making maltose more available. This is particularly significant as maltose and its derivatives have been linked to adhesion in other organisms. For example, maltose has been shown to inhibit adhesion and biofilm formation in *Pseudomonas aeruginosa* and enhance adhesion to epithelial cells in *Candida albicans* [39,40]. In fact, the deletion of an *Hbt. salinarum trmB* homolog has been linked to reduced protein glycosylation, an interesting observation in the context of the growing understanding that glycosylation affects pilin functionality [19,41].

Lastly, *hvo_2734*, encoding a Gcn5-related N-acetyltransferase (GNAT), a universal protein family that uses acyl-CoA to acylate a wide variety of substrates, is of particular interest since few of these proteins have been functionally characterized thus far and their potential role in facilitating surface adhesion may provide much-needed insight [42].

## 4. Conclusions

While archaea are mostly known for inhabiting extreme environments, they are ubiquitous and can be found in habitats ranging from garden soil to the human microbiome. However, compared to eukaryotes and bacteria, the other two domains of life, we still know relatively little about archaea. Our studies have clearly underscored this reality, as only a small subset of the identified genes had a previously known function in motility or adhesion. Future characterization of these highly diverse genes and their relevant operons, and further screening, will likely result in significantly better understanding of the mechanisms facilitating and regulating motility and adhesion. In addition, these mutants may help delineate the roles that these processes play in scavenging nutrients and adapting to stress.

## Figures and Tables

**Figure 1 life-06-00041-f001:**
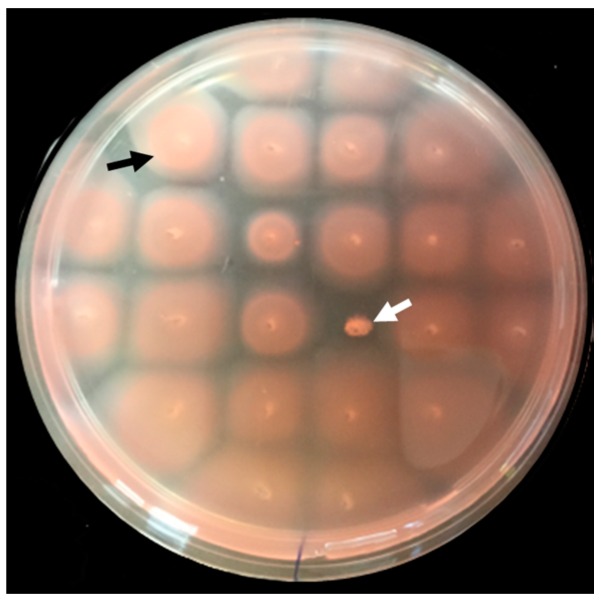
Motility screen using semi-solid agar plates allowed for the isolation of non-motile *Hfx. volcanii* mutants. Cells from 23 colonies isolated from a *Hfx. volcanii* transposon library, together with a wild type control strain (FH37), were stab-inoculated into a motility plate (0.3% agar) and incubated at 45 °C for five days. The control strain and one motility mutant are indicated with a black and a white arrow, respectively.

**Figure 2 life-06-00041-f002:**
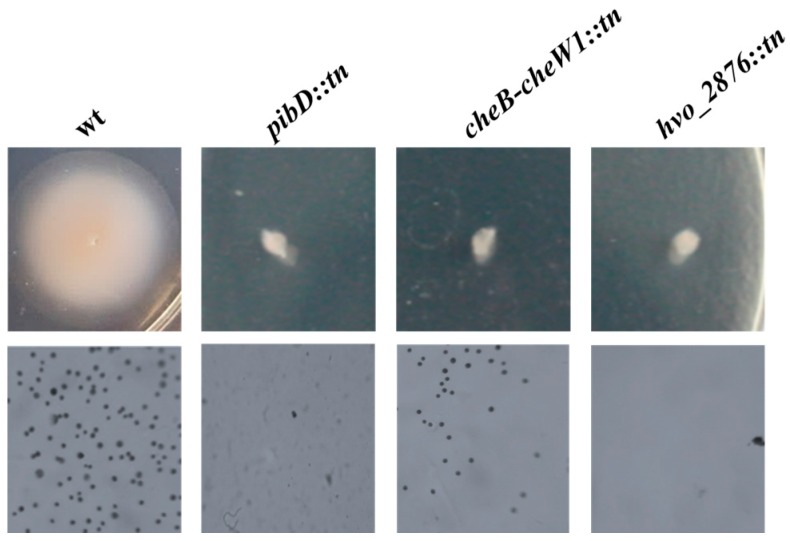
A subset of transposon insertion motility mutants also exhibits an adhesion defect. Motility and surface adhesion assays of *Hfx. volcanii* FH37 (wt), and the transposon mutants *pibD::tn*, *cheB-cheW1::tn*, and *hvo_2876::tn*. Motility was tested by stab-inoculating cells into motility plates and was incubated at 45 °C for three days (Bars 10 mm) (top). Adhesion to plastic coverslips was tested using a modified ALI assay [18]. Coverslips were placed in individual wells of 12-well plates, each containing 3 mL of a mid-log phase liquid culture. After overnight incubation, cells were fixed with 2% acetic acid, stained with 0.1% crystal violet, and observed by light microscopy (1000× magnification) (*n* = 3) (bottom).

**Figure 3 life-06-00041-f003:**
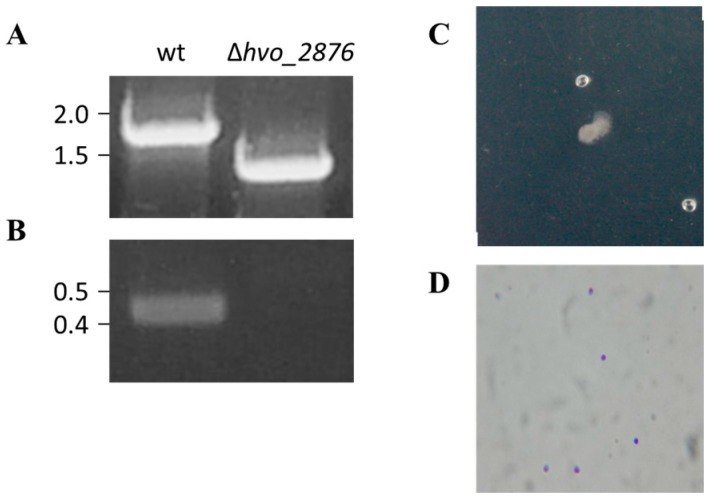
Characterization of Δ*hvo_2876*. PCR amplification was performed using primers (**A**) against the flanking regions located approximately 700 bp upstream and 700 bp downstream of *hvo_2876*, and (**B**) specific for the *hvo_2876* gene. The template DNA used was isolated from the wild-type or Δ*hvo_2876* strain. (**C**) Motility and (**D**) adhesion assays using Δ*hvo_2876* were carried out as described in Figure 2.

**Figure 4 life-06-00041-f004:**
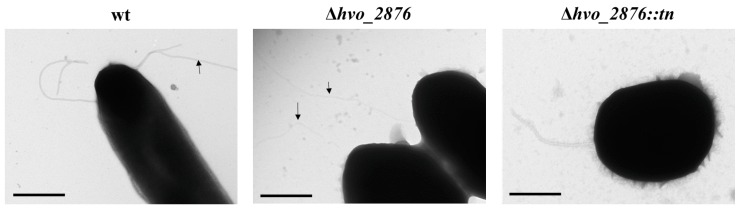
Cells lacking *Hvo_2876* contain fewer surface filaments. TEM of whole cells of wild-type, Δ*hvo_2876*, and *hvo_2876::tn* strains. This image of wild-type cells represents about 40% of all wild-type cells analyzed, while the images of Δ*hvo_2876* cells with long filaments represents about 20% of either *hvo_2876::tn* or Δ*hvo_2876* cells. Between 20% and 30% of fixed *hvo_2876::tn* or Δ*hvo_2876* cells contained short filamentous structures, while only about 10% of fixed wild-type cells have such structures. Bar: 500 nm.

**Figure 5 life-06-00041-f005:**
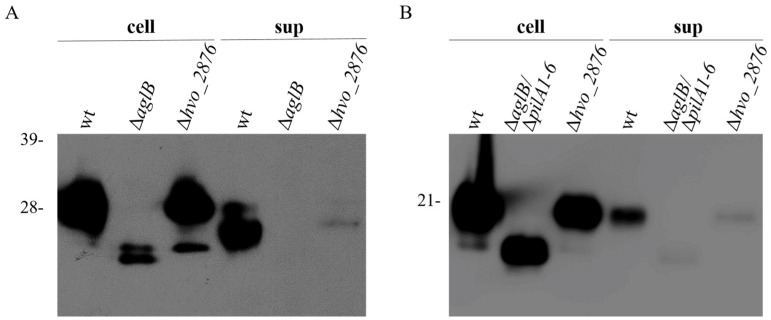
A Δ*hvo_2876* strain has fewer flagella or pili than a wild-type strain. Western blot analysis using anti-His antibodies was performed on protein extracts isolated from cell lysates (cell), and on TCA-precipitated proteins from the supernatants (sup) of wild-type, Δ*aglB*∆*pilA1–6*, or Δ*hvo_2876* liquid cultures expressing either (**A**) FlgA1His or (**B**) PilA2His. Molecular mass standards are indicated on the left (in kDa).

**Figure 6 life-06-00041-f006:**
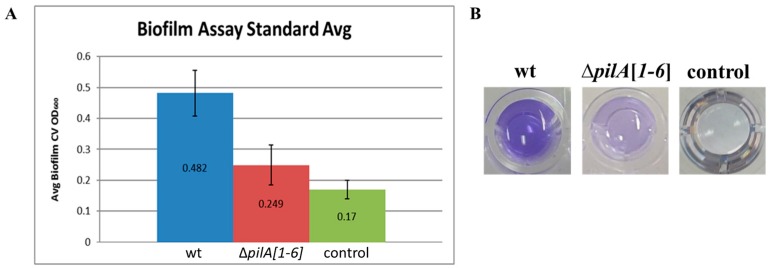
A modified 96-well ALI assay can distinguish between normal and deficient surface adhesion. (**A**) Averages ± standard deviations of eight isolates of wild-type (FH37) and Δ*pilA[1–6]* strains along with eight negative controls of CA liquid media alone were graphed to compare adhesion by quantification of crystal violet (CV) using optical density (OD_600_). (**B**) Images of released crystal violet in wells. Darker purple hue represents more adhering cells present.

**Figure 7 life-06-00041-f007:**
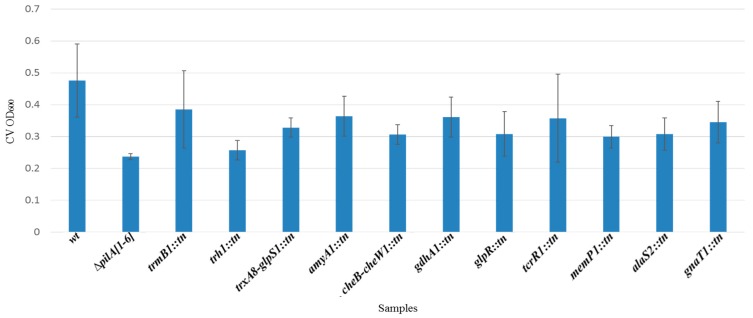
The adhesion deficiency of transposon mutants quantified by measuring the optical density (OD_600_) of crystal violet (CV). A 96-well ALI assay was performed as described in Materials and Methods. Absorption averages ± standard deviations of three isolates each for wt (FH37), Δ*pilA[1–6]*, and 11 transposon mutant strains were plotted.

**Figure 8 life-06-00041-f008:**
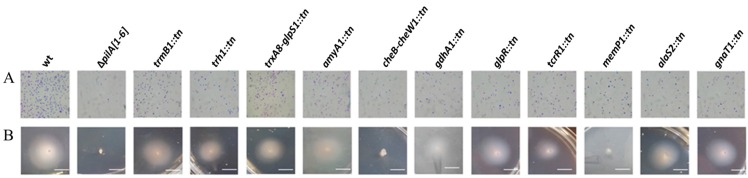
A subset of transposon insertion adhesion mutants also exhibit severe motility defects. (**A**) Adhesion of wt (FH37), Δ*pilA[1–6]*, or transposon mutants to plastic coverslips was determined as described in Figure 2 and used to confirm the adhesion phenotypes observed in 96-well plates; (**B**) Motility assays using the same strains were performed as described in Figure 2. Bars, 10 mm.

**Table 1 life-06-00041-t001:** Plasmids and Strains.

Plasmid or Strain	Relevant Characteristic(s)	Reference or Source
Plasmids		
pTA131	Amp ^r^; *pyrE2* under a ferredoxin promoter	[24]
pTA963	Amp ^r^; *pyrE2* and *hdrB* markers, inducible *ptna* promoter	[25]
pJS141	pTA963 containing 6X-His-tag	This study
pMT9	pTA963 containing *FlgA1His*	[18]
pMT24	pTA963 containing *pilA1His*	[26]
pRE3	pTA963 containing *pilA2His*	[26]
pEY2	pTA131 containing Δ*hvo_2876*	This study
pEY3	pJS141 containing *hvo_2876*	This study
*E. coli* strains		
DH5alpha	F-80d*lacZ*ΔM15 Δ(*lacZYA-argF*)*U169 recA1 endA hsdR17*(rK- mK-) *supE44 thi-1 gyrA relA1*	Invitrogen
DL739	MC4100 *recA dam- 13::*tn9	[27]
*H. volcanii* strains		
H53	Δ*pyrE2* Δ*trpA*	[24]
FH37	H53 containing *trpA* inserted between *hvo_2072* and *hvo_2073*	[28]
EY2	FH37 containing pEY02	This study
EY3	FH37 containing pMT9	This study
EY4	FH37 containing pMT24	This study
EY5	FH37 containing pRE3	This study
EY6	FH37 containing pEY3	This study
RE43	H53 Δ*pilA1* Δ*pilA2* Δ*pilA3* Δ*pilA4* Δ*pilA5* Δ*pilA6*	[26]

**Table 2 life-06-00041-t002:** Primers used for PCR amplification.

Primer Name	Sequence (5’–3’)	Target Sequence
Fwko2876A	AAGCTAGGATCCAGAACCTTCCCCGGC	700 bp upstream of *hvo_2876* start codon
HVO2876KOXbaRv	GAACCTTCTAGAATCGGCATCGAGCCG	700 bp downstream of *hvo_2876* stop
codon Fwoverlap2876	GACGAATCCACGCCTGCGTCGAACGCCGAG	15 bp upstream of stop codon
Rvoverlap2876A	CTCGGCGTTCGACGCAGGCGTGGATTCGTC	15 bp downstream of start codon
HVO2876CompFW	GATATCCATATGGCAACCTCGACG	At the start codon
HVO2876CompRV	GATATCGAATTCGGCGACGAGCTTGAAGTACAGC	At the stop codon

**Table 3 life-06-00041-t003:** *Hfx. volcanii* transposon motility mutants.

Genomic Location	Transposon Disrupted Gene Product	Upstream Gene Product	Downstream Gene Product
*hvo_0069* {71094}	Sulfatase	Pan E (2-dehydropantoate 2-reductase)	Hypothetical protein
*hvo_0246* {223313}	ArsR family transcription regulator	Phosphodiesterase	Hypothetical protein
*hvo_0246* {223129}	ArsR family transcription regulator	Phosphodiesterase	Hypothetical protein
*hvo_0448 ^* {397819}	Imidazoleglycerol-phosphate synthase subunit HisH	Phosphate-binding protein	PheA1 (Prephenate dehydratase)
*hvo_0558* {493398}	Adenylyltransferase	Htr15a (transducer protein Htr15) ^#^	Sulfurtransferase
*hvo_1098* {1004042}	LysA (diaminopimelate decarboxylase)	DapF (diaminopimelate epimerase)	DapD (2,3,4,5-tetrahydropyridine-2,6-dicarboxylate N-succinyltransferase)
*hvo_1224/25^* {1116022}	CheB (protein-glutamate methylesterase) and CheW1 (purine-binding taxis protein)	CheA (taxis sensor histidine kinase) ^#^	Putative sugar transporter ^#^
*hvo_1224/25^* {1115464}	CheB (protein-glutamate methylesterase) and CheW1 (purine-binding chemotaxis protein)	CheA (taxis sensor histidine kinase) ^#^	Putative sugar transporter ^#^
*hvo_1374* {1252418}	Long chain fatty acid-CoA ligase	Acd4 (acyl-CoA dehydrogenase)	MenE (2-succinylbenzoate-CoA ligase)
*hvo_1376* {1254749}	Methylmalonyl-CoA epimerase	MenE (O-succinylbenzoate-CoA ligase)	UPF0145 family protein^#^
*hvo_2876* {2712944}	Hypothetical protein	Probable S-adenosylmethionine-dependent methyltransferase *^#^	Putative sugar transporter
*hvo_2993* {2823760}	PibD (prepilin/preflagellin peptidase)	HisI (phosphoribosyl-AMP cyclohydrolase) ^#^	Hypothetical protein *
*hvo_2993* {2823079}	PibD (prepilin/preflagellin peptidase)	HisI (phosphoribosyl-AMP cyclohydrolase) ^#^	Hypothetical protein *
*hvo_3001* {2833319}	ABC transporter permease	LigA DNA ligase (NAD(+))	ABC transporter ATP-binding protein
*hvo_3001* {2833976}	ABC transporter permease	LigA DNA ligase (NAD(+))	ABC transporter ATP-binding protein
*hvo_3001* {2833517}	ABC transporter permease	LigA DNA ligase (NAD(+))	ABC transporter ATP-binding protein
*hvo_3002* {2834820}	ABC transporter ATP-binding protein	ABC transporter permease **	Hypothetical protein

Annotations from NCBI unless stated otherwise. {} transposon insertion site. ^ transposon insertion just outside a gene or in between genes. * gene is within 30 nucleotides of the gene the transposon inserted into. ** gene overlaps with the gene the transposon inserted into. ^#^ annotation from Halolex, UC Davis.

**Table 4 life-06-00041-t004:** *Hfx. volcanii* adhesion mutants identified using a modified ALI assay.

Genomic Location	Transposon Disrupted Gene Product	Upstream Gene Product	Downstream Gene Product
*hvo_0169* {154195}	TrmB family transcriptional regulator [TrmB1]	EmrE (small multidrug export protein) ^#^	Hjc (Holliday junction resolvase) ^#^
*hvo_0179* {162826}	Trh1 (transcriptional regulator)	Hypothetical protein	FtsJ (23S rRNA (uridine-2'-O-) methyltransferase) ^#^
*hvo_0543/44 ^* {474633}	TrxA8 (Thioredoxin) and galactose proton symporter [GlpS1]	Hypothetical protein	Hypothetical protein
*hvo_0567* {503715}	AmyA1 (alpha-amylase)	Glucan 14-alpha-glucosidase	TrmB family transcription regulator ^#^
*hvo_1224/25^* {1116027}	CheB (protein-glutamate methylesterase) and CheW1 (purine-binding taxis protein)	CheA (taxis sensor histidine kinase) ^#^	Putative sugar transporter
*hvo_1451* {132276}	GdhA1 (glutamate dehydrogenase)	MaoC (molybdenum cofactor biosynthesis protein)	Citrate lyase
*hvo_1501* {1369466}	GlpR (DeoR-type DNA-binding transcriptional regulator)	PfkA (1-phosphofructokinase) *	LeuB (3-isopropylmalate dehydrogenase)
*hvo_2447* {2315733}	Two component response regulator [TcrR1]	DMT superfamily transport protein *^#^	Probable oxidoreductase ^#^
*hvo_2512* {2378600}	Membrane protein [MemP1]	GatD (glutamyl-tRNA(Gln) amidotransferase subunit D)	Hypothetical protein
*hvo_2717* {2562834}	AlaS2 (alanyl-tRNA synthetase)	Acd5 (Acyl-CoA dehydrogenase)	Cro/C1 family transcription regulator ^#^
*hvo_2734* {2580223}	GNAT acetyltransferase [GnaT1] ^#^	Hypothetical protein	Hypothetical protein

Annotations from NCBI unless stated otherwise. {} transposon insertion site. [] unofficial protein abbreviations only used for the purposes of this paper. ^ transposon inserted between genes. * gene is within 30 nucleotides of the gene the transposon inserted into. ^#^ annotation from Halolex, UC Davis.

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
