# Peer review of "Screening of a Haloferax volcanii Transposon Library Reveals Novel Motility and Adhesion Mutants"

_life, 2016, doi:10.3390/life6040041_

Reviewer 1 Report

In this report, Legerme and colleagues describe the use of a transposon-based screen to define novel Haloferax volcanii genes involved in motility and adhesion. In doing so, they have identified new targets that may serve in these processes in future studies.

Before manuscript can be accepted, the following points require attention:

Introduction –The introduction is a jumble of statements looking at mechanism, biosynthesis and transcriptional regulation of flagella and pili in two archaeal phyla. It is very difficult to follow the flow of logic. Please rewrite in a more organized and coherent manner. In rewriting, add background for people not in the field. For instance, not everyone knows what are euryarchaeota and crenarchaeota or that pilins are the building blocks of pili. Also, end the text found in the first two paragraphs with some direction as to where this article is heading. As now written, the reader is left with the idea that we are going to learn something new about N-glycosylation.

Tables 3 and 4 – Are any of these genes adjacent to other genes that might also be involved in the processes being studied, pointing to operons or gene clusters?

Line 272 – Can the anti-His antibodies be used in EM studies?

Figure 5a – The blot reveals staining of more than one band in each of lanes 2-4. What are these? Indeed, the major band is such a blob, it is hard to say anything about its size.

Line 307 – What is the ALI assay? Give a brief description.

Figure 8 – The picture frame cuts the upper labels in half. Expand the frame.

Line 368 – The Schmid lab has linked TrmB to N-glycosylation in Hbt. salinarum (Todor et al., 2014). Could this be relevant here?

Author Response

See attached PDF.

Reviewer 2 Report

Find enclosed my comments about the article “Screening of a Haloferax volcanii transposon library 2 reveals novel motility and adhesion mutants” from the authors Georgio Legerme, Evan Yang, Rianne N. Esquivel, Saija Kiljunen, Harri Savilahti and 4 Mechthild Pohlschroder. I enjoy reading this article. This is a well-written article, the assays are well design and results are interesting. However, the genomic data are poorly described. I suggest the category of “Accepted after major revision”

Major considerations:

The genomic sequencing material an methods is missing too many details: For intance, what was utilized to purified the bacterial chromosome? How the library was made?

Where are the bioinformatics analysis? What pipeline was utilized for the assemble and annotation?

Regarding the results of the bioinformatics analysis, I did not see the NCBI accession number for the sequenced strain. What was the coverage? How many rRNA? Why the author did no perform a phylogenic analysis? In other words, I did not see any results from the sequencing. Perhaps is convenient to remove the sequencing form this article.

Minor suggestions

Line 39: the term “prokaryotic” is incorrect. Please use bacteria, archea or eukaryote

Line 108: Define NZCYM and their components

Line 153: Give a reference for LDS. Why instead of SDS?

Line 160: MOPS concentration?

Author Response

See attached PDF.

Round  2

Reviewer 2 Report

I am satisfied with the authors modifications